# Validity of a Single Inertial Measurement Unit to Measure Hip Range of Motion During Gait in Patients Undergoing Total Hip Arthroplasty

**DOI:** 10.3390/s25113363

**Published:** 2025-05-27

**Authors:** Noor Alalem, Xavier Gasparutto, Kevin Rose-Dulcina, Peter DiGiovanni, Didier Hannouche, Stéphane Armand

**Affiliations:** 1Kinesiology Laboratory, Geneva University Hospitals and University of Geneva, 1205 Geneva, Switzerland; 2Center of Research on Skeletal Muscle and Movement, Geneva University Hospitals and University of Geneva, 1205 Geneva, Switzerland; 3Division of Orthopaedics and Traumatology, Faculty of Medicine, Geneva University Hospitals, 1205 Geneva, Switzerland

**Keywords:** total hip arthroplasty, hip range of motion, inertial measurement units, machine learning, artificial neural networks

## Abstract

Hip flexion range of motion (ROM) during gait is an important surgery outcome for patients undergoing total hip arthroplasty (THA) that could help patient monitoring and rehabilitation. To allow systematic measurements during patients’ clinical pathways, hip ROM measurement should be as simple and cheap as possible to ensure patient and clinician acceptance. Single IMU options can match these requirements and offer measurements both during daily living conditions and standardized clinical tests (e.g., 10 m walk, timed up-and-go). However, single-IMU approaches to measure hip ROM have been limited. Thus, the objective of this study was to explore the accuracy of one IMU in measuring hip ROM during gait and to determine whether a single-IMU approach can provide results comparable to those of multi-IMU systems. To assess this, machine learning models were employed, ranging from the simplest (linear regression) to more complex approaches (artificial neural networks). Eighteen patients undergoing THA and seven controls were measured using a 3D opto-electronic motion capture system and one thigh-mounted IMU. Hip ROM was predicted from thigh ROM using regression and classification models and was compared to the reference hip ROM. Multiple regression was the best-performing model, with limits of agreement (LoA) of ±13° and a systematic bias of 0. Random forest, RNN, GRU and LSTM models yielded LoA ranges > 27.8°, exceeding the threshold of acceptable error. These results showed that one IMU can measure hip ROM with errors comparable to those of two-IMU methods, with potential for improvement. Using multiple linear regression was sufficient and more appropriate than employing complex ANN models. This approach offers simplicity and acceptance to users in clinical settings.

## 1. Introduction

Osteoarthritis (OA) is a degenerative joint disease with increasing prevalence and risk factors, such as obesity and low physical activity. It is characterized by joint pain, swelling and activity limitations [1]. For patients with end-stage hip OA, total hip arthroplasty (THA) is a common, successful and cost-effective surgical solution to relieve pain and restore function. While pain reduction post-surgery is well handled, functional improvement remains a key research goal for enhancing patient quality of life [2]. Patients show functional improvement to about 80% of the control group level, with remaining deficits in terms of gait speed and hip flexion ROM during gait [3,4].

The objective measurement of hip ROM during gait is commonly used to assess hip function before and after surgery [3]. When assessing function, the World Health Organization’s International Classification of Function defined two domains of function: ‘Capacity’ and ‘Performance’. Capacity is what an individual can accomplish in a standardized setting, while performance is what an individual accomplishes during daily living activities (ADL). The assessment of these two fields and their interactions is recommended to fully assess patient function.

Hip ROM is currently mostly measured by using three-dimensional (3D) marker-based optoelectronic motion capture (OMC). Although OMC is known for its accuracy and for being a gold standard in measuring gait parameters in clinical/laboratory settings [5], it is hindered by the need for expensive and bulky equipment, a large laboratory space and complex preparations (e.g., precise placement of markers on anatomical landmarks of participants, camera calibration and proper lighting) and is limited to capacity assessment. Such constraints lead to limited applicability in the daily clinical routine, where simplicity is key for both patient compliance and implementation by healthcare teams [6].

Despite well-known limitations when compared to OMC [7,8], inertial measurement units (IMUs) offer several practical advantages that show their high potential for measuring functional outcomes in clinical applications: fast measurements, simple setups, cost-effectiveness and usability in both clinical and ecological settings. This last point is especially relevant as it can provide measures of both functional capacity and performance with the same device, allowing for direct comparisons between laboratory-based and real-world (ADL) functions. This would enable a clear evaluation of both domains and their interaction [9]. Moreover, recent research suggests that remote functional assessments could serve as outcomes in clinical trials and provide additional information to standardized assessments for informing clinical decision making [10].

IMUs have been proposed for quantifying joint kinematics and ROM, including hip kinematics in patients after THA using two IMUs [11]. However, the use of two IMUs has shown limited accuracy, with previous studies reporting limit of agreement ranges between 15° and 30° [11,12]. Moreover, an important drawback of a multi-IMU system is low patient compliance due to the discomfort of wearing multiple sensors and the complexity of the setup. Single IMU solutions offer simplicity, cost-efficiency and time-efficiency [13], making them an ideal candidate for clinical application. A single IMU solution to measure hip flexion ROM would provide a tool for daily use in clinics to assess and follow function in large cohorts of patients.

Given this perspective, several single-IMU studies were conducted to measure gait parameters in daily living conditions [9,13,14,15,16], such as cadence in patients with cerebral palsy [17] or walking speed in patients with various pathologies [13]. Two prior studies focused on measuring lower-limb kinematics using one IMU. Bonnet et al. measured sagittal hip joint angles using N-harmonic Fourier series modeling and a least squares approach during static flexion–extension exercises for rehabilitation [18], while Sung et al. estimated hip joint angles based on a long short-term memory (LSTM) neural network trained on 30 healthy participants [19]. Both studies provided encouraging results, with a root mean square difference (RMSD) of 3.5° and a correlation of 0.9 for the first [18] and a root mean square error (RMSE) below 7° with a coefficient of determination (R^2^) of 0.74 for the second [19]. Among the few gait analysis studies on patients undergoing THA, those that aimed to quantify hip sagittal angles and ROM used a minimum of two sensors [11,12]. A study conducted by Zügner et al. measured hip flexion–extension angles during gait with LoAs of [−10, 5°] [11]. The study by Piche et al. estimated ROM at three different speeds (0.8, 1.2 and 1.6 m/s) during treadmill gait and reported respective LoAs between 15° and 30° [12].

Machine learning (ML) has been shown to be a relevant approach for kinematics or ROM prediction by potentially reducing computation time and wearable sensor quantity [20,21]. The applications of ML (i.e., clustering, regression, classification) have recently been expanding in movement biomechanics to estimate complex relationships between input data, such as kinematic waveforms, and a given output in numeric or class format (e.g., kinematics or disease status) [22]. Regression models capture the relationship between wearable sensor inputs and biomechanical outputs, where one study employed ML regression models to predict hip and knee joint angles during rehabilitation exercises after THA and TKA using a single IMU [23]. The current artificial neural network (ANN)-based methods of joint angle assessment usually require a multiple IMU system where one IMU is attached to each segment [11,12,13,14,15]. Mundt et al. predicted lower limb joint angles during gait using ANNs and obtained RMSE values less than 4° in all planes [24]. Tan et al. predicted knee flexion for patients with knee OA using raw IMU data measured from thigh and shank segments and reported RMSE values between 7 and 11.8° [25]. Despite increased research in the estimation of joint angles and ROM with the application of ML methods, no previous study has measured hip ROM during gait using a single IMU in patients undergoing THA. There is thus a lack of focus on measuring hip ROM using a single-IMU setup, despite its great interest to clinicians.

The aim of this study was to validate the measure of sagittal hip ROM in patients undergoing THA with a single IMU, combined with ML models of various complexity levels, and to assess whether its performance is comparable to that of multi-IMU systems. We hypothesized that one thigh IMU is sufficient to estimate hip ROM using ML models. Since a standard difference (S.D.) below 5° on joint angles is considered acceptable for clinical interpretation with OMC-based Clinical Gait Analysis [26], and as we used Bland Altman LoAs to validate the prediction accuracy, the validation criterion was a LoA range < 27.8° between the predicted and reference hip ROM. To be representative of future clinical use, this evaluation was performed in patients undergoing THA, both before and after surgery.

## 2. Materials and Methods

### 2.1. Participants

This study included 18 patients (sex = 61.1% F, age = 62.0 ± 8.3 years old, weight = 73.5 ± 12.7 kg, height = 170.5 ± 9.9 cm) undergoing THA and 7 asymptomatic controls (sex = 85.7% F, age = 67.7 years old, weight = 64.2 ± 13.8 kg, height = 64.2 ± 13.8 cm).

The patients attended three visits: before THA (n = 18), 3 months after THA (n = 12) and 1 year after THA (n = 8). The control group attended only one visit.

### 2.2. Equipment

The participants were simultaneously measured with one IMU on each thigh (Physilog6, MindMaze, Lausanne, Switzerland) and a twelve-camera OMC system (Oqus7+, Qualysis, Göteborg, Sweden). Clusters of 4 markers were fixed to the IMUs to track their orientation in the same frame as the OMC, providing a ground truth for IMU orientation. The IMUs were then placed on the lateral side of each thigh (Figure 1). These positions were selected to ensure minimal artifacts caused by soft tissue [27]. Double-sided adhesive tape and elastic bands (SuperWrap straps, Qualysis, Göteborg, Sweden) were used to firmly attach the IMUs and clusters to the skin of the participants. The three-dimensional acceleration and angular velocity were acquired at 128 Hz, with ranges of ±16 g and ±2000°/s, respectively. Reflective markers were attached to the participants’ skin according to the Conventional Gait Model (CGM) 1.0 [28]. Marker trajectories were measured at 100 Hz. Gait trials were acquired from the participants using the OMC and IMU systems, posteriorly synchronized together.

### 2.3. Protocol

To synchronize both systems, at the start and end of each session, all IMUs were fixed and aligned in a custom-made rectangular plate equipped with 4 reflective markers (Figure 1), and ten rotations were performed and measured by OMC and IMUs. Cross-correlation was performed between the angular velocity measured by the IMUs and by the OMC to assess delay at the start and end. Since the delay between the OMC and IMUs was found to increase linearly with time, a linear fit was performed to obtain the delay between the systems as a function of measurement time.

During each visit, the participants were asked to perform gait trials along a 10 m straight path, including 10 at self-selected speed, 2 at slow speed and 4 at fastest speed.

### 2.4. Pre-Processing

Marker trajectories during gait were labeled with QTM software (Qualisys, version 2023.3), and gait events were computed automatically from the markers [29]. A manual check was performed afterward to ensure correctly labeled events. The IMU 3D angular velocity data were low-pass-filtered using a finite impulse response filter (2nd-order, 3.2 Hz) [17]. The filter order was chosen to efficiently attenuate the undesired frequencies, and the cutoff frequency was chosen to permit the detection of fast gait cycles. The filter was applied to the angular velocity data twice (filtfilt, MATLAB R2023a) to obtain zero-phase distortion.

### 2.5. Data Processing

Data processing was performed using MATLAB R2023a software (Mathworks, Natick, MA, USA). The processing workflow for computing thigh and hip ROM is presented in Figure 2 and detailed in the following subsections.

#### 2.5.1. Cluster Orientations

The marker positions of the cluster were designed to match the axis of the IMU fixed to it [27]. The coordinate system (CS) and pose for each thigh cluster were computed directly from the 3D positions of the markers in the OMC frame.

#### 2.5.2. Sensor-to-Segment Calibration

Sensor-to-segment alignment was performed through manual–functional calibration [8]. The long axis of the IMUs was aligned with the thigh axis. The average flexion–extension axis of the thigh was determined with a principal component analysis (PCA) on the 3D angular velocity during gait trials and was used as the medio-lateral axis of the thigh [27]. The direction of the vector was determined from the sign of the PCA vector component. Cluster-to-segment calibration was performed using the same method to ensure consistency between the IMU and cluster CS definitions.

#### 2.5.3. Kinematics Computation

IMU ROM Computation:

Multiple methods were tested and validated against OMC (thigh cluster). These methods included strapdown integration as well as Kalman and complementary filters [7] (Table A1, Appendix A). The method with the lowest error when compared to the thigh cluster ROM was used for prediction.

To implement strapdown integration, the IMU thigh flexion angle was computed for each gait cycle, and trapezoidal integration was applied on the filtered medio-lateral component of angular velocity of the thigh (acquired at 128 Hz). IMU ROM was then simply the range of this thigh angle. To implement Kalman and complementary filter integration, sensor fusion algorithms were tested, and IMU Euler angles (ZYX sequence) were calculated from the orientation data represented in quaternions. All sensor fusion codes were found available on GitHub (https://github.com/marcocaruso/sensor_fusion_algorithm_codes, accessed on 27 March 2021) from [7]. IMU ROM was then calculated from the range of the IMU thigh Euler angles.

Hip ROM Computation:

The reference hip joint kinematics were computed according to the CGM 1.0 convention [28]. Hip ROM, measured with CGM 1.0, was used as the ground truth to evaluate the accuracy of the IMU-based ROM measurements.

### 2.6. ML Models

Several models were employed in this study to estimate hip ROM. The below-listed models were tested for two distinct goals: regression to predict hip ROM values, and classification to predict hip ROM classes (Figure 3). As their name suggests, the simple and multiple linear regression models were used only for regression, while all other models were used for both regression and classification. However, the outputs of regression models were also classified and visualized in confusion matrices to evaluate their potential to classify ROM. The same pipeline was followed for both types of ML/RNN models. The only difference between the regression and classification is that the output of classifiers was hip ROM class, where the ROM values were classified into 3 categories: reduced, average and normal hip ROM. This class division was based on one-third of the distribution of hip ROM across all participants to maintain the maximum number of samples possible for each class to reduce class imbalance (Figure A1). To evaluate the overall performance of the naïve and ML models, a 5-fold group cross-validation was performed, treating each participant with all their measured visits (pre-surgery and 3 months and 1 year post-surgery) along with every limb side as a single group (e.g., Participant 1—Left).

Before employing regression models, a naïve model was used with the mean of reference hip ROM values as the predicted value. This was performed to evaluate the potential of the other regression models (Section 2.6.1, Section 2.6.2, Section 2.6.3 and Section 2.7) in outperforming the naïve model.

#### 2.6.1. Simple Linear Regression (SLR)

A simple linear regression between the thigh ROM and the hip ROM was performed with the ‘fitlm’ function in MATLAB R2023a.

#### 2.6.2. Multiple Linear Regression (MLR)

A multiple linear regression model (Equation (1)) was tested by introducing additional features to the IMU ROM in the model. A selection of features to be used as predictors was performed based on a multiple correlation analysis (Pearson’s R) to assess their importance and avoid information redundancy. Features with a Pearson’s R ≥ 0.5, indicating strong correlation based on Cohen’s Guidelines [30], were dropped if they showed high correlation with another feature. Therefore, among each set of 2 correlated features (R ≥ 0.5), the feature with the higher correlation (R) with hip ROM was selected. To further refine feature selection, a regression decision tree was used for predictor importance analysis, and the features with the highest importance were selected using the ‘*predictorImportance*’ function in MATLAB R2023a [31]. The equation used for MLR is shown below.(1)$$Y={ϐ}_{0}+{ϐ}_{1}X_{1}+{ϐ}_{2}X_{2}+\ldots +\;{ϐ}_{n}X_{n}$$

#### 2.6.3. Random Forest

Random forest (RF) models consist of a set of decision trees that grow in randomly selected subspaces of data. They are fast, easy to implement and among the most accurate general-purpose learning techniques available, providing non-linear multiple regression [32]. These models are defined by two hyperparameters: the number of decision trees and the minimum number of leaves per tree. These parameters are predetermined before the training takes place. The RF model used in this study was trained with IMU ROM and the same set of features selected in Section 2.6.2 as inputs.

### 2.7. Network Architecture

Deep RNN models are neural networks used to process sequential or time-series data. The RNNs selected for hip ROM prediction are known to be suitable for time-series analysis and have been previously trained for joint motion prediction: Simple RNN [33], Gated Recurrent Unit (GRU) [34] and long short-term memory (LSTM) networks [25,35]. In our study, RNNs were created via a subject-general approach for predicting hip ROM, where three different RNNs were trained. These models are defined by three main hyperparameters: the learning rate, the dropout rate and the number of units for each recurrent layer.

The architecture of each model included an input layer with a length of 100, corresponding to that of a gait cycle, and 7 channel arrays, corresponding to the number of inputs. The input data were standardized by removing the mean and scaling them to unit variance using the ‘StandardScaler’ function in Python. Two recurrent layers, with the first being bidirectional, were included [25]. Two hidden layers were added following the recurrent layers, and a dropout layer was added after each recurrent and hidden layer. The hidden and recurrent layers were created by hyperbolic tangent functions (tanh) for regression [36] and by linear rectified units (ReLus) for classification [37]. An output layer was created by a linear activation function to return the estimated hip ROM value after regression, and the function ‘softmax’ [38] was used to return the hip ROM class after classification. The loss functions were the mean squared error between the predicted and reference ROM for regression [22] and the sparse categorical cross-entropy for classification [39]. The models were trained using the adaptive momentum (Adam) optimization algorithm [38]. The data processing, machine model and experimental results of the neural networks were developed and implemented using Python 3.12.0 with the libraries Pandas, Numpy, Scikit learn, Keras and TensorFlow.

#### 2.7.1. Hyperparameter Tuning

To optimize the accuracy of the machine learning models (RF and RNNs), the hyperparameters were fine-tuned and optimized by a grid search across multiple training and testing iterations to identify the optimal model configuration, which was subsequently used for analysis. The criterion for hyperparameter tuning was based on IMU time-series data (IMU sagittal ROM, 3D acceleration and 3D angular velocity) as the input and hip ROM as the output, and the loss functions were the mean squared error and sparse categorical cross-entropy for the regression and classification models, respectively.

#### 2.7.2. Model Training

Nested cross-validation (Figure 4) was used to identify the best model with the best hyperparameters because of its effectiveness for small datasets, despite being computationally expensive [40]. Therefore, outer and inner cross-validation loops for RNNs were automated using Python to find the optimal set of parameters with respect to the accuracy metric across all inner loops. Outer cross-validation was performed to report the average metrics for each fold and across all folds for all networks. Four inner folds were used for hyperparameter tuning, and one held-out fold was reserved for testing. This approach ensured that the model was evaluated on unseen data during the hyperparameter tuning process.

For classification, the validation process was performed in a stratified manner to maintain the percentage of samples for each class as much as possible while adhering to the constraint of non-overlapping groups. This was performed using the ‘StratifiedGroupKFold’ function in Python. The accuracy, F1-score and confusion matrices were used as evaluation metrics for each model [41]. The execution time taken for each model was measured for a 13th Gen Intel^®^ Core™ i9-13900 @ 2 GHz (Intel Corporation, Santa Clara, CA, USA) during the nested cross-validation process of RNNs.

### 2.8. Statistical Analysis

#### 2.8.1. Normality and Significance Analysis of SFA Errors

The Shapiro–Wilk test was used to test the normality of the error distribution among SFAs, and the Wilcoxon rank sum test was used for statistical significance analysis between SFA errors (*p* < 0.05).

#### 2.8.2. Evaluation of Validity

The evaluation metrics for the regression were the MAE, LoA and R^2^. The systematic bias, LoAs and standard difference were evaluated with Bland–Altman analysis to assess the agreement between the reference and predicted ROM. First, the agreement between the IMU ROM and the hip ROM was evaluated to define the baseline level of agreement. Then, all models were evaluated with the same method.

Based on this study’s hypothesis, an S.D. < 5° was considered clinically acceptable. Therefore, a predicted hip ROM with a LoA range of 27.8° was considered acceptable [42].

The evaluation metrics for classification were the accuracy and F1-score percentages.

## 3. Results

### 3.1. Feature Selection

The extracted features are shown in Figure 5. The features selected using multiple correlation analysis were the thigh ROM, walking speed (computed using OMC) and participant leg length, as well as the angular velocity mean, skewness, crest factor and minimum valley calculated per gait cycle.

After further refining the selection process using feature importance analysis (Figure 6), the features selected were the thigh ROM, leg length and walking speed, as they showed the highest contribution rankings. These features are biomechanical parameters that can physically explain the variance in hip ROM; greater IMU ROM, leg length or walking speed lead to greater hip ROM.

### 3.2. Regression

Figure 7 summarizes the Bland–Altman plots of agreement for each trained regression model, where the multiple linear regression shows the lowest LoA range and, thus, the best prediction performance compared to the other models. A detailed demonstration of every model’s Bland–Altman individual plot is shown in Figure A3 (Appendix A), where every point in each plot represents one gait cycle of a participant.

Before prediction, and using only thigh ROM as the independent variable, the bias was −8.4°, with a LoA range of 30.5° and a linear pattern of increasing difference as hip ROM increased (Figure 8). The multiple linear regression (MLR) performed best in terms of LoA and systematic bias, while the RNN, GRU and LSTM models performed worse than the simple and multiple linear regression. MLR led to a bias of 0 and a LoA range of 26.8° (Figure 9), below the validity level of 27.8°. MAE and R^2^ values are reported in Appendix A (Table A2).

### 3.3. Classification

To evaluate the performance of the classifiers and the ability of the regressors to distinguish classes, confusion matrices were used (Figure 10) as well as accuracy and F1-score metrics. Despite the lower error (LoA range) of the SLR and MLR models compared to the RF or RNN models, there was still confusion between the classes because of an imbalance in the class distribution. This explains why class 1, i.e., the class with the least observations (Figure A1, Appendix A), showed the lowest percentage of correctly predicted classes in all the models. The accuracy and F1-score of the classifiers did not exceed 0.6% (Table A3, Appendix A), indicating poor classification performance. The model with the highest accuracy and F1-score was the RF classifier, with the least computation time (480 s) after a five-fold cross-validation. It is worth noting that the lower the complexity and computation time of the models, the better the performance obtained, with F1-scores of 0.58, 0.47, 0.41 and 0.35 for the RF, RNN, GRU and LSTM classifiers, respectively.

## 4. Discussion

The aim of this study was to explore the accuracy of a single thigh IMU in measuring hip ROM during gait and to determine which model was the most accurate for such prediction. For the MLR and RF regression models, the IMU ROM, angular velocity time-domain and participant biomechanical features were used as input. As for the RNNs (regressors and classifiers), only IMU ROM and raw data were used as input because of the models’ known potential to analyze time-series data [43].

This study, to our knowledge, is the first to assess the accuracy of a single IMU in measuring hip ROM during gait in patients undergoing THA. This is also the first study that attempts to classify hip ROM based on raw data from a thigh-attached IMU. The best performing ML algorithm was MLR, resulting in an MAE of 4.2 ± 0.5°, LoAs of ±13° and a systematic bias of 0. These results indicate that this single-IMU approach is comparable to those using multiple IMUs and predicts with similar error. Using a multi-IMU setup, Piche et al. estimated ROM at three different speeds (0.8, 1.2 and 1.6 m/s) during treadmill gait and reported respective biases and LoA ranges (°) of 5.19 (−4.93, 15.30), 5.95 (−2.28, 14.19) and 4.84 (−9.99, 19.67) [12]. Another two-sensor method, conducted by Zügner et al., measured gait hip flexion–extension angles with LoAs of [−10, 5°] and a systematic bias of −3° [11]. Considering the MAE metric, the MAE of 4.2° obtained in this study is lower than that reported by Renani et al., who obtained an RMSE of 7.2° using a multi-IMU system and deep learning models to predict hip flexion–extension angles [44]. Our best model’s MAE is comparable to that of Hernandez et al., who reported a similar average MAE of 3.6° on peak hip flexion prediction using convolutional and LSTM layers [45]; however, their result was obtained using a five-IMU system’s raw data collected from the pelvis, thighs and shanks. They obtained low MAE values of 3.7° and 3.8° for the right and left hips, respectively, with a good correlation of > 0.8 to reference (OMC) hip angles. However, since they did not report Bland–Altman LoA for peak flexion values, we cannot assume that the estimation of peak hip flexion is accurate and had no systematic bias in their study.

This study’s best model resulted in a Pearson’s correlation coefficient (R) of 0.69, indicating a good correlation between ROM values [30] but a low level of variance (R^2^ = 0.43) explained by the used inputs, showing poor model fitting and the need for improvement. Testing higher complexity models, like RF, RNN, GRU and LSTM, did not seem to improve prediction, where the LoA range increased with increased complexity. This indicates that multiple regression, with its lower complexity, outperformed the RF and RNN models in predicting ROM. It is worth noting that while the simpler ML models (SLR, MLR and RF) outperformed the naïve approach (Figure A2, Appendix A), the more complex models (Figure A3, Appendix A) did not achieve better performance than the naïve model.

Previous studies have examined the accuracy of ML models in predicting hip joint kinematics or ROM using a minimum of two IMUs and their raw acceleration and angular velocity data [23,24,35,43,45,46,47], and one study added anthropometric data to the input raw data [22]. Mundt et al. reported their prediction accuracy using normalized RMSE (nRMSE = 9.8%) over the data range, which is lower than our RMSE of 13.6% [43]. Argent et al. explored several ML regression models to predict hip and knee joint angles using only a thigh and shin IMU for each joint, respectively [23]. Although they obtained an average RMSE of 4.99°, they only measured joint angles during certain rehabilitation movements rather than during gait. Lim et al. attempted to measure lower-limb joint angles using the center of mass of an IMU fixed to the lower back of healthy participants [48], but they only obtained the angles of the thighs, shanks and feet as output, making it difficult to compare our results to theirs.

In a clinical setting, user acceptance is essential when using a wearable measurement system. The LoA range of difference for hip ROM measurement obtained by MLR in this study, using only one IMU, is acceptable, showing that an easy implementation of a single-IMU system is possible in clinical settings, reducing patient burden and minimizing the time required for practitioners to set up the system. We chose to measure gait hip ROM in THA patients as it is the most frequent activity performed in their everyday life. Since hip ROM measurement using one IMU is reasonably valid, further work should be performed to assess its reliability before its usage as a clinical outcome or for rehabilitation [49]. Therefore, we measured gait in the laboratory using IMU and OMC (reference) systems to provide a simple, fast and accurate capacity tool that could be used in clinics. In this regard, this study also attempted to classify hip ROM based on raw data from a thigh-attached IMU in patients undergoing THA. The aim of this method was to create a simple and objective scale based on IMU data that clinicians could use to assess hip function, like the commonly used subjective scales [50]. Among all classifiers, the RF model obtained the highest accuracy (0.59%) based on a five-fold nested cross-validation and performed fastest during computation (480 s). This shows that the model is not accurate in classifying hip ROM, especially looking at the confusion matrix (Figure 9), with a higher confusion between classes 1 and 2. This can be explained by the fact that class 1 has the lowest number of samples in our dataset, with around 200 values of ROM, compared to classes 2 and 3, with samples of around 300 and 400. Since no prior work has assessed the accuracy of hip ROM classification using one IMU, our results cannot be compared to the literature.

This study has some limitations. First, the sample size of 25 participants is relatively small for an ML-based model to be trained and evaluated for prediction performance. This represents a large limitation of this study, as the robustness of the relationship between input and output variables in an ML model relies heavily on the quantity of training data [51]. On another note, the lack of sufficient data leaves us questioning the source of the obtained errors, where the source and solution of the linear trend observed in all Bland–Altman plots remains unclear. In other words, we cannot deduce whether the errors resulted from sensor misalignment, soft tissue artifacts or a random source. Concerning sensor drift as another source of error, we assume that integrating IMU angular velocity signals for the computation of ROM was not affected by drift, as the integration was performed over short-duration signals (i.e., gait cycles). Second, the alignment of the IMU to the thigh segment was based on a functional calibration using PCA under the assumption that the gait of the participants was straight. However, this was not the case for all the participants. In this study, the LoA range obtained by MLR was below the acceptable range, showing that a single-IMU method is as accurate as previously mentioned multiple-IMU methods [11,12]. A third limitation is the imbalance in the percentage of participants by sex, given that 61.1% and 85.7% of the THA patients and healthy controls were female, respectively. This results in a bias in the dataset; however, it is important to note that females account for a higher percentage in the hip OA population [1], and the recruitment of the female controls was conducted to match the sex distribution of the patients. Another limitation concerning the classification models is class imbalance, since the participants were not evenly distributed across the three classes of hip ROM. This explains the low accuracy and F1-score that the models yielded. Stratified cross-validation was applied to tackle this limitation; nevertheless, it still did not fully compensate for the lack of samples per class due to the relatively low observations in the whole dataset.

## 5. Conclusions

This study showed that a single IMU on the thigh, combined with a multiple linear regression model, could estimate hip ROM with acceptable accuracy and comparable validity to multi-IMU setups. The results showed that the use of a low-complexity regression model outperformed more complex ML models in predicting hip ROM. Accurately predicting hip ROM using one sensor can facilitate the measurement of this key functional outcome in ADL, especially since it improves user acceptance. Our results demonstrate the feasibility of this approach using regression with an acceptable error, but they also highlight the potential for further improvements to enhance prediction accuracy.

## Figures and Tables

**Figure 1 sensors-25-03363-f001:**
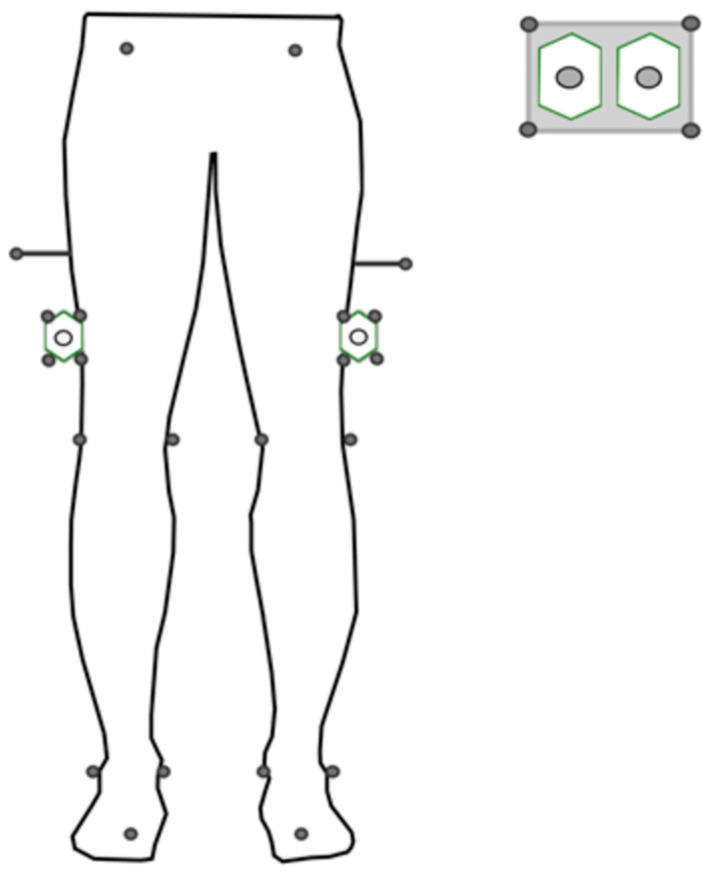
Protocol setup, including 2 IMUs (Physilog6, GaitUp) and reflective markers on clusters and anatomical landmarks.

**Figure 2 sensors-25-03363-f002:**
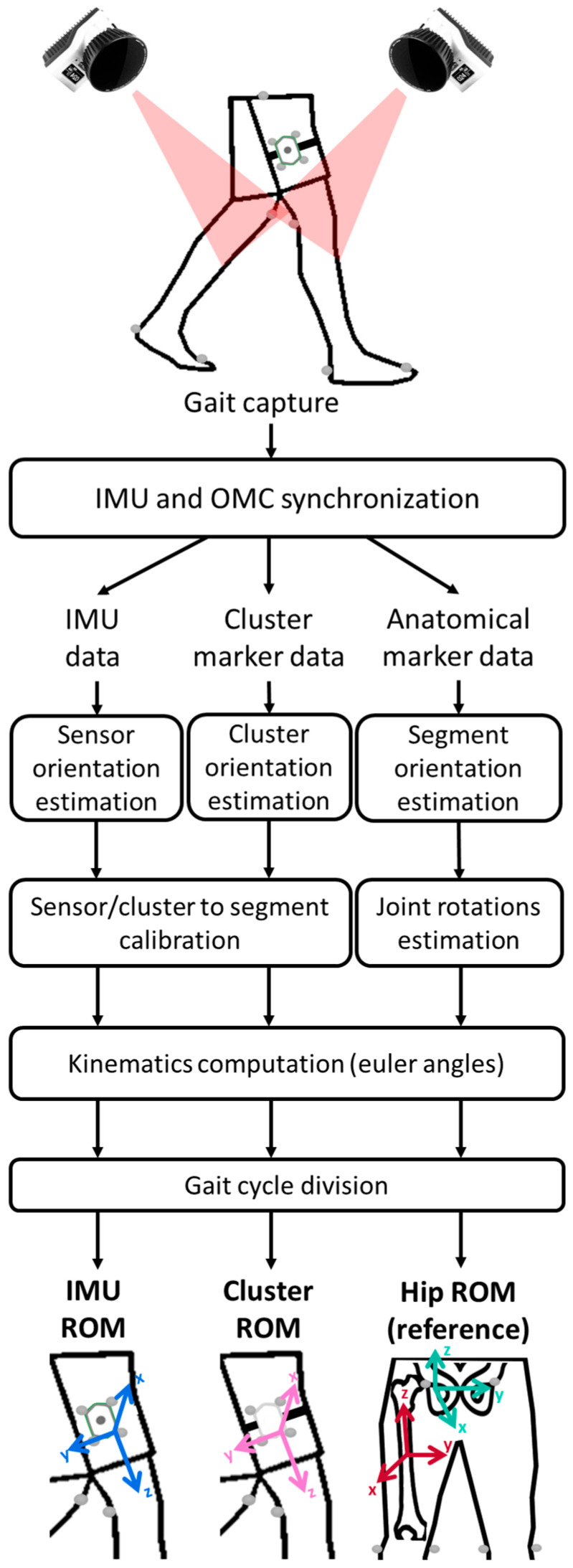
Data acquisition and processing steps for thigh and hip ROM computation.

**Figure 3 sensors-25-03363-f003:**
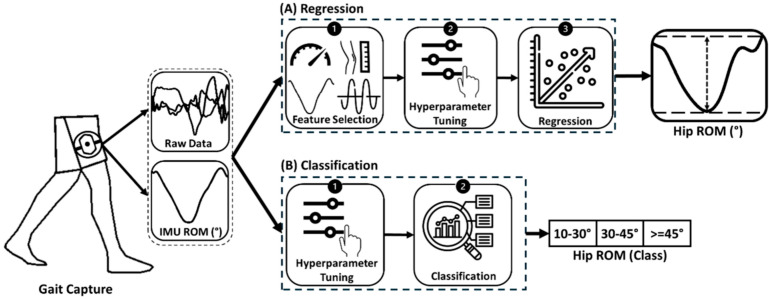
ROM prediction design. (**A**) Regression: IMU ROM and a set of features/raw data were selected as the input, and the output was the hip ROM value. (**B**) Classification: IMU ROM and raw data were selected as the input, and the output included three classes of hip ROM (reduced, average and normal). Nested cross-validation, including hyperparameter tuning, was performed in both pipelines to obtain the best model with the most suitable parameters.

**Figure 4 sensors-25-03363-f004:**
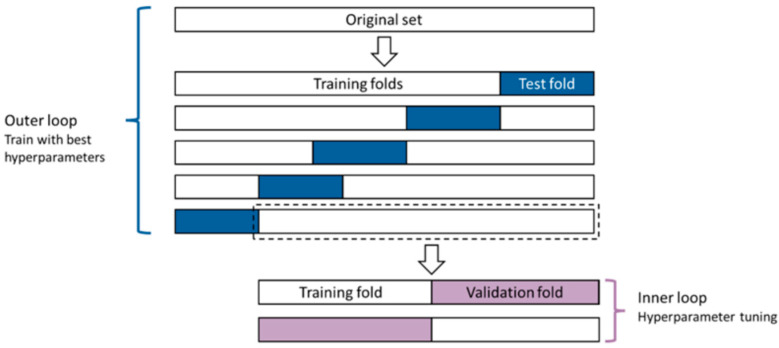
Nested cross-validation for model selection.

**Figure 5 sensors-25-03363-f005:**
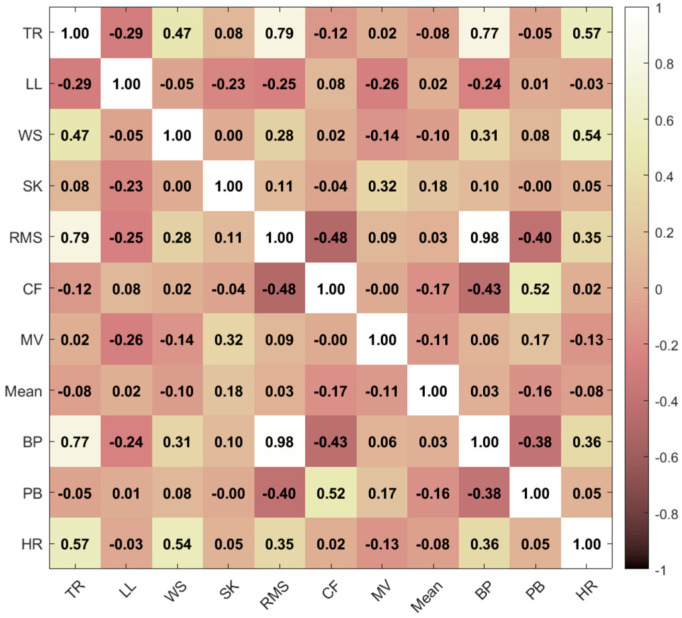
Multiple correlation analysis for feature selection. Features that had a high (R ≥ 0.5) correlation with other features were eliminated. Abbreviations: TR: thigh ROM; LL: leg length; WS: walking speed; SK: skewness; RMS: root mean square; CF: crest factor; MV: minimum valley; BP: band power; PB: power bandwidth; HR: hip ROM.

**Figure 6 sensors-25-03363-f006:**
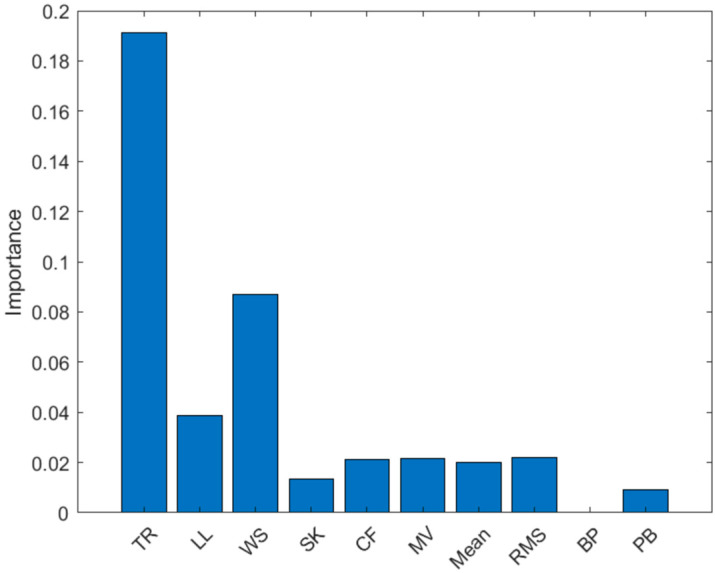
Ranking of the 10 extracted features by importance.

**Figure 7 sensors-25-03363-f007:**
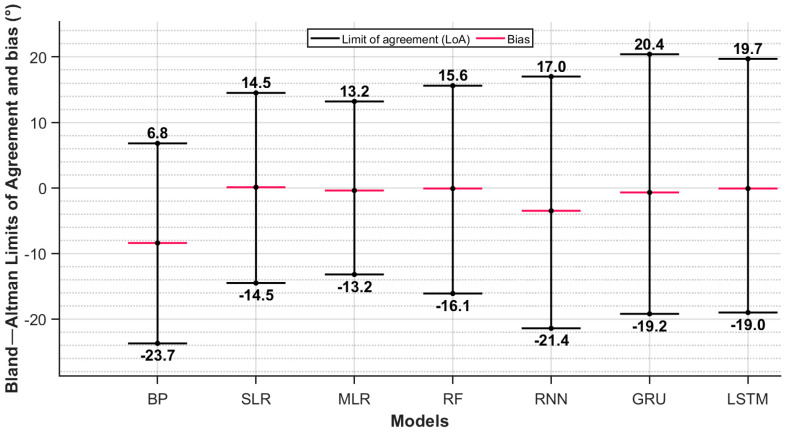
Bland–Altman limits of agreement (LoA) and mean biases before and after hip ROM prediction using different regression models. The systematic difference in each model is as follows: Before prediction (BP): -8.4°, SLR: 0.1°, Multiple Linear Regression (MLR): 0, random forest (RF): −0.1°, Recurrent Neural Network (RNN): −3.5°, and Gated Recurrent Units (GRU): −0.7°, Long Short-Term Memory (LSTM): −0.1°.

**Figure 8 sensors-25-03363-f008:**
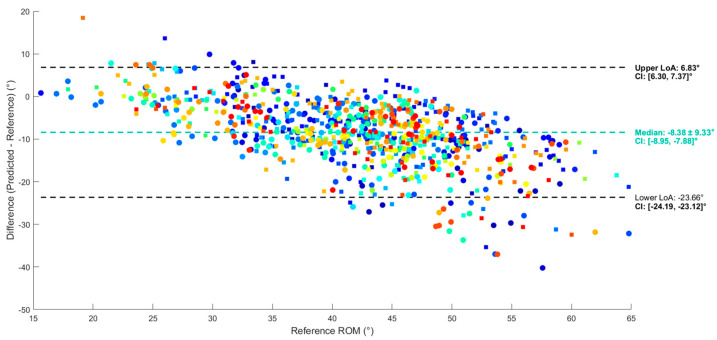
Bland–Altman plot of the agreement between IMU and hip (reference) ROM before regression. Each point corresponds to a gait cycle from a participant.

**Figure 9 sensors-25-03363-f009:**
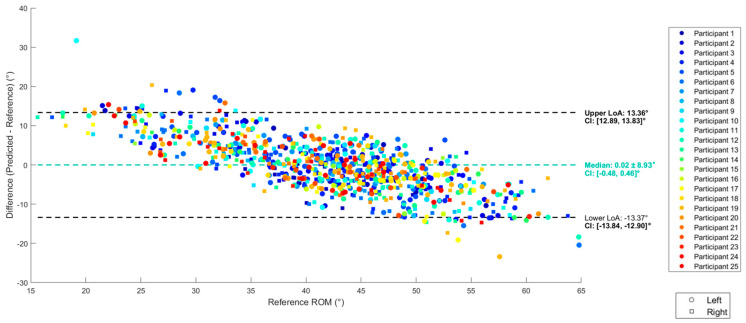
Bland–Altman plot of agreement between the best model’s (MLR) predicted ROM and reference hip ROM.

**Figure 10 sensors-25-03363-f010:**
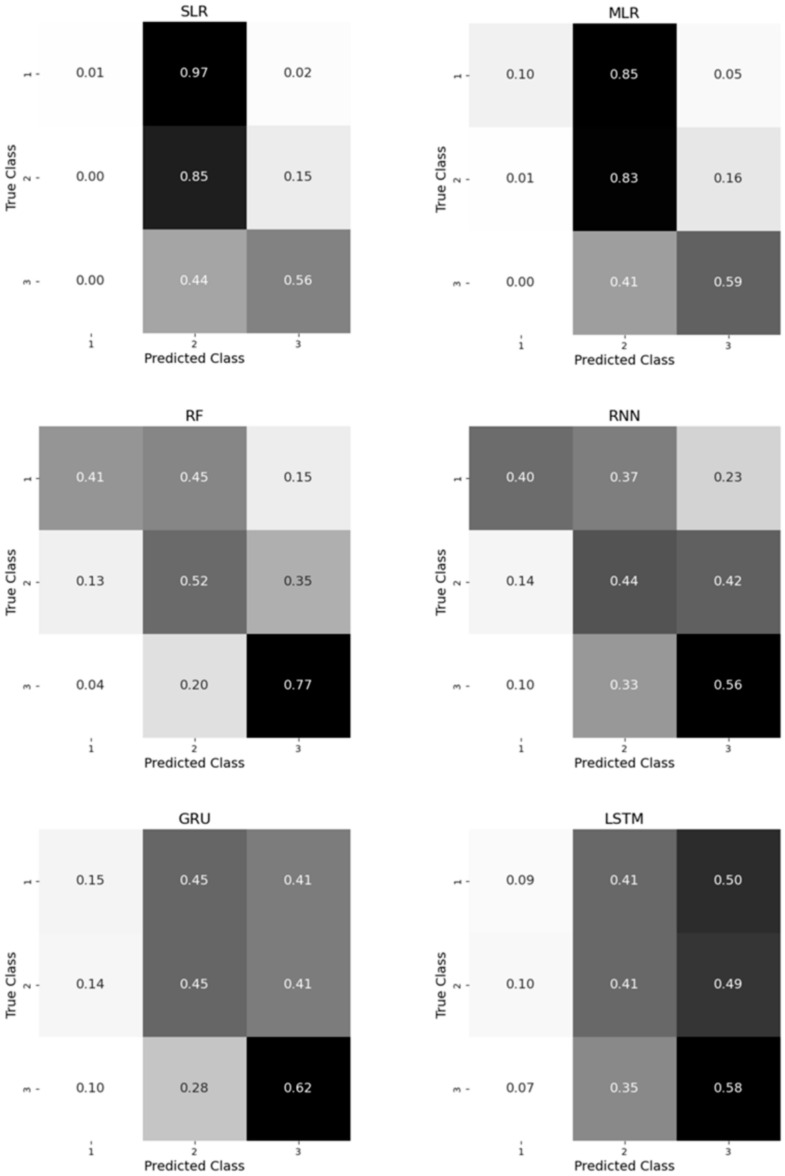
Confusion matrices of hip ROM classifiers for three classes of ROM: reduced (10–30°) (1), average (30–45°) (2) and normal (≥45°) (3).

## Data Availability

Data sharing is not applicable to this article.

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
