# Peer review of "Validity of a Single Inertial Measurement Unit to Measure Hip Range of Motion During Gait in Patients Undergoing Total Hip Arthroplasty"

_sensors, 2025, doi:10.3390/s25113363_

Round 1

Reviewer 1 Report

Comments and Suggestions for Authors

The authors mention that there have been other ML-based single IMU studies for other joint angles. Could it be reasonably assumed that, because those methods worked for other joint angles that they would naturally work for hip ROM as well? Is there a specific reason why hip ROM is so much more unique and nuanced that it motivated the authors to do this study?

Could the authors comment on whether or not an unsupervised model would be appropriate here? I'm curious how the model developed in the work will translate to less controlled clinical settings where the markers and positioning of the IMU may not be as well controlled.

Figure 5 is very blurry.

Author Response

Dear Reviewer,

Thank you very much for your valuable comments. Please find attached our response in a word document. 

Kind Regards,

Reviewer 2 Report

Comments and Suggestions for Authors
  1. In the multiple linear regression (MLR) analysis, feature selection was performed by eliminating variables with Pearson's R ≥ 0.5. Could the authors  consider using more robust feature selection methodologies for further optimize model performance?
  2. In the conclusion, the authors claim that he underperformance of more complex ML model compared to the low complexity regression model is due to the insufficient amount of data volume. Was there a systematic investigation into specific causes of potential overfitting in complex models? For instance, were significant discrepancies observed between training error and validation error during model evaluation?
  3. While the IMU placement on the lateral thigh was intended to minimize soft tissue artifacts, such artifacts might remain substantial in patient populations with elevated BMI. Was the variation in IMU measurement errors systematically evaluated across different BMI groups?
  4. The IMU fixation method employed double-sided adhesive tape and elastic bands. Was the impact of alternative fixation approaches on data acquisition stability assessed? 

Author Response

(The authors gave the same response as above.)

Reviewer 3 Report

Comments and Suggestions for Authors

This paper considers using a single IMU for hip ROM measurement in patients undergoing total hip arthroplasty, and emphasizes that hip flexion range of motion (ROM) during gait is an important surgery outcome for patients undergoing total hip arthroplasty (THA) that could help patient monitoring and rehabilitation. Perhaps this is an interesting topic. However, the paper needs to be further improved and clarified in the following issues:
1) The title is presented in the form of a rhetorical question, which is not particularly suitable for academic writing. Moreover, it fails to effectively convey the key aspects of the research and its innovative elements. It is advisable to revise the title to better reflect the study's content and contributions.

2)The abstract section is expected to comprehensively incorporate the challenging questions, along with the innovative solutions proposed and the corresponding experimental results obtained. This would ensure that the abstract effectively summarizes the core aspects of the study, providing readers with a clear and concise overview of its significance, approach, and outcomes.

3)The introduction section needs to provide the advantages and disadvantages of existing research methods, the problems to be solved, the motivation of this paper's research, and finally, the contributions of this paper need to be condensed point by point.

4)In section 2.5 Data Processing, details such as how to obtain the data, what the data format is, and the size of the data, etc., need to be provided.

5)In section 2.6 ML Models, the deep learning network model needs to be provided, the mathematical theory of the model needs to be analyzed in detail, the network loss function is essential, and the network training results and performance need to provide corresponding curves.

6)In section 3. Results, it is essential to provide the patient's on-site experimental situation. Additionally, the study should include visualized representations of the experimental results, as these graphical depictions can enhance the clarity and comprehensibility of the findings. 

In addition, experimental comparisons with the state-of-the-art methods are also required.

Author Response

(The authors gave the same response as above.)

Reviewer 4 Report

Comments and Suggestions for Authors

This paper focuses on the accuracy of using a single Inertial Measurement Unit (IMU) to measure the hip range of motion (ROM) in patients undergoing total hip arthroplasty. Overall, the research has certain application value, but there are still some areas for improvement. My concerns are as follows.

  1. Supplement more relevant research on IMU measurement in gait - type diseases, such as "Global joint information extraction convolution neural network for Parkinson’s disease diagnosis" and "Intelligent iot anklets for monitoring the assessment of parkinson’s diseases".
  2. Describe in detail the specific types of hip diseases in patients, the grading of disease severity and other information.
  3. Explain the reasons for choosing a specific low - pass filter (2nd order, 3.2Hz).
  4. Explain in detail the theoretical relationships between the selected features (such as walking speed, leg length, etc.) and hip ROM, and how these features affect model prediction. Adopt methods such as feature importance analysis to show the contribution of each feature to the model results.
  5. Further discuss the specific guiding significance of the research results for clinical practice, such as how to develop personalized rehabilitation programs based on the measurement results of a single IMU.
  6. There are issues with excessive conceptual explanations and verbose language in the paper.

Author Response

(The authors gave the same response as above.)

Round 2

Reviewer 3 Report

Comments and Suggestions for Authors

No more comments.

Reviewer 4 Report

Comments and Suggestions for Authors

accept